# Advances in Understanding the Acyl-CoA-Binding Protein in Plants, Mammals, Yeast, and Filamentous Fungi

**DOI:** 10.3390/jof6010034

**Published:** 2020-03-10

**Authors:** Shangkun Qiu, Bin Zeng

**Affiliations:** JiangXi Province Key Laboratory of Bioprocess Engineering, College of Life Sciences, Jiangxi Science & Technology Normal University, Nanchang 330013, China; 15738826862@163.com

**Keywords:** Acyl-CoA-binding protein, structure, function, plants, mammals, yeast, filamentous fungi

## Abstract

Acyl-CoA-binding protein (ACBP) is an important protein with a size of about 10 kDa. It has a high binding affinity for C_12_–C_22_ acyl-CoA esters and participates in lipid metabolism. ACBP and its family of proteins have been found in all eukaryotes and some prokaryotes. Studies have described the function and structure of ACBP family proteins in mammals (such as humans and mice), plants (such as *Oryza sativa*, *Arabidopsis thaliana*, and *Hevea brasiliensis*) and yeast. However, little information on the structure and function of the proteins in filamentous fungi has been reported. This article concentrates on recent advances in the research of the ACBP family proteins in plants and mammals, especially in yeast, filamentous fungi (such as *Monascus ruber* and *Aspergillus oryzae*), and fungal pathogens (*Aspergillus flavus*, *Cryptococcus neoformans*). Furthermore, we discuss some problems in the field, summarize the binding characteristics of the ACBP family proteins in filamentous fungi and yeast, and consider the future of ACBP development.

## 1. Introduction

Acyl-CoA-binding protein (ACBP) was first discovered in the brain of mice in 1983 and was initially named diazepam-binding inhibitor (DBI) [1]. Subsequently, ACBP and its family of proteins were found in other eukaryotes, such as animals, plants, fungi, and protozoa, as well as prokaryotes, including pathogens of some animals and plants [2]. ACBP is approximately 10 kDa. The ACBP family proteins all contain a highly conserved ACBP domain, which plays an important role in acyl-CoA binding and transport. Some ACBP family proteins also contain other domains, such as the ankyrin repeat (ANK) domain, kelch motif, transmembrane domain, ECH domain, and GOLD domain [3,4]. ACBP is a key lipid transport protein and plays an important role in intracellular lipid synthesis, vesicle transport, and gene regulation [5,6]. ACBP has a high affinity and specificity for long-chain acyl-CoA esters and medium-chain acyl-CoA esters, and ACBP can protect acyl-CoA esters from hydrolysis during some physiological activities, such as lipid biosynthesis and remodeling, β-oxidation, and protein acylation [2].

Recently, research on the structure and function of ACBP has mainly focused on mammals and plants. Some studies have shown that mammalian ACBP is mainly located in the cytoplasm, endoplasmic reticulum (ER), Golgi body, and nucleus [7]. Through the deletion of the ACBP gene in mice, researchers found that the biosynthesis of fat in hepatocytes was delayed when the mice were weaned. Therefore, this indicates that ACBP plays an important role in intracellular lipid synthesis and metabolism [8]. By knocking out ACBP in mice, researchers found that the content of ceramide in the hepatocytes of mutant mice decreased significantly [9]. Recent studies have shown that there are six ACBP family proteins in *Hevea brasiliensis*, among which HbACBP1 and HbACBP2 play an important role during lipid metabolism and latex synthesis [10]. 

At present, research on the structure and function of ACBP family proteins in yeast and filamentous fungi is increasing, especially the important role they play in lipid metabolism. Studies have shown that the ACBP from yeast delays the growth of yeast and increases the biosynthesis of sphingolipids [11]. By deleting the Acb1 gene, which encodes ACBP in *Saccharomyces cerevisiae*, researchers demonstrated that ACBP has an important function in cell vacuole accumulation and maintaining the cell membrane structure [12]. Some studies have recently shown that overexpressing ACBP5 in *Monascus ruber* increased the production of Monascus pigments (MPs), the water-soluble segment, and the ethanol-soluble segment by 11.67%, 9.8%, and 12.70%, respectively [13]. ACBP is known as AoACBP in *Aspergillus oryzae*. Using fluorescence co-localization, studies have demonstrated that the punctate structures of AoACBP are distributed in the cytoplasm, and punctate structures exhibited microtubule-dependent motility, revealing the potential mechanism of AoACBP’s involvement in lipid metabolism in *A. oryzae* [14]. Therefore, it is of great significance to study the action mechanism of the ACBP family proteins in organisms, especially to explain the important mechanism of the ACBP family proteins involved in lipid metabolism in yeast and filamentous fungi. 

## 2. General Characteristics of ACBP Structure and Function

The ACBP family proteins which contain the ACBP domain are also named ACBD, and DBI is named ACBD1 [15]. Different ACBP homologous proteins are found in different species, which contain different numbers of ACBP. For example, there are seven kinds of ACBD family proteins in humans, among which the shortest protein containing only the ACBP domain is called ACBD1 or ACBP. ACBD5 contains 525 amino acid residues and only the ACBP domain. The other five ACBD family proteins contain not only the ACBP domain but also other domains, such as the ANK domain, GOLD domain, and ECH domain [15]. The yeast *Saccharomyces carlsbergensis* contains two ACBP family proteins, known as yeast ACBP types 1 and 2, which exhibit 48% (type 1) and 49% (type 2) conservation of amino acid residues with the human ACBP [16]. *Caenorhabditis elegans* contains seven ACBP family proteins, among which ACBP-1, 3, 4, 6 only contain the ACBP domain [17]. There are six ACBP family proteins in *Arabidopsis thaliana*, which are called AtACBP-1, 2, 3, 4, 5, 6. The ACBP family proteins in *A. thaliana* are the most studied plant ACBP family proteins [15]. 

ACBP family proteins of plants can be divided into four categories according to the different domains they contain [18,19]. The first class is ACBP, which has a single ACBP domain and a small molecular weight, usually consisting of 88–155 amino acids. The second class, ANK ACBP, is usually composed of 260–370 amino acids. In addition to the ACBP domain, this ACBP family protein contains an ANK domain at the C-terminus of the protein. The third class is a large molecular weight ACBP that has a single ACBP domain, which is usually composed of 215–700 amino acids. The fourth category is the kelch motif ACBP, which typically consists of 648 to 668 amino acids and contains a kelch domain at the C-terminus along with the ACBP domain [10,20]. In addition to the kelch domain and the ANK domain, ACBP family proteins in humans and mammals also contain different numbers of the ECH domain, GOLD domain, and Herpes DNAp domain [15]. Different domains have different functions. For example, ACBP interacts with other proteins through the kelch motifs and ANK domain [19]. The conserved ACBP domain of the ACBP family proteins binds with very high affinity to ligands of medium- and long-chain acyl-CoA esters [21]. ACBP binds to the ER and cell membrane through the transmembrane domain and the ANK domain [3,22] (Figure 1).

Different ACBP family proteins have different functions. For example, researchers performed a comparative lipidome analysis of 613 lipid molecules from the liver, muscle, and plasma of weaning and adult *ACBP* knockout and wild type mice. The results show that the deficiency of ACBP affects lipid metabolism in the liver and plasma during weaning [5]. Other studies have shown that Aichi viruses replicate by interacting with ACBD3 in host cells, and the deficiency of ACBD3 in host cells can inhibit the proliferation of Aichi viruses [23]. Hepatitis C virus is a pathogen that causes many chronic liver diseases, cirrhosis, and even liver cancer. Researchers knocked out the *ACBD3* gene in the host cells, and the results showed that the protein expression level of the Hepatitis C virus was significantly elevated, suggesting that ACBD3 inhibits viral replication [24]. ACBP from *A. thaliana* is involved in lipid metabolism and can maintain a membrane-associated acyl pool during lipid transport from the ER to the plasma membrane [20]. Some studies have shown that ACBP can affect spore formation by regulating lipid metabolism. Researchers deleted AcbA from *Dictyostelium discoideum* and showed that *D. discoideum* produced fewer spores compared to the wild type [25].

## 3. The Function of ACBP Family Proteins in Plants

### 3.1. ACBP Family Proteins Can Promote Latex Formation

*H. brasiliensis* is a critical cash crop capable of synthesizing high-quality latex, which is commercially important [26]. Recently, studies have shown that *H. brasiliensis* contains six ACBP family proteins that are divided into four categories. The first category, HbACBP1, only contains the ACBP domain. The second category, HbACBP2, not only contains ACBP domain but also an ANK domain. The third category, composed of HbACBP3 and HbACBP4, contains only ACBP domains. However, the number of amino acids is significantly more than HbACBP1. The fourth category, composed of HbACBP4 and HbACBP5, contains not only an ACBP domain but also a kelch domain. Through RT-qPCR analysis of the expression patterns of these six ACBP family proteins, researchers found that HbACBP1 and HbACBP2 were mainly expressed in latex, and their expression levels were significantly higher than in other organs and tissues. HbACBP3-6 are mainly expressed in mature leaves and the male flower, indicating that they may play a critical role in the growth and development of mature leaves and male flower, while HbACBP1 and HbACBP2 mainly play an important role during latex formation. The expression patterns of ACBP family proteins in laticifers were analyzed using three methods: bark tapping (a mechanical ground), exogeneous ethrel (ET, a releaser of ethylene), and jasmonic acid (JA) stimulation. The results showed that bark tapping and JA stimulation increased the expression levels of HbACBP1 and HbACBP2, whereas ET only increased the expression level of HbACBP1 [10]. ET and bark tapping can promote the production of latex [27,28], and JA stimulation can promote laticifer differentiation [29], indicating that HbACBP1 and HbACBP2 play important roles in latex formation and lipid metabolism [10]. Therefore, as an important commercial resource, it is of great significance to explore the role of ACBP family proteins during the formation of latex.

### 3.2. OsACBP1 and OsACBP2 Are Functionally Distinct in Oryza sativa

*O. sativa* (rice) contains six homologous ACBP family proteins known as OsACBP1,2,3,4,5,6 [18]. OsACBP1 and OsACBP2 play important roles in intracellular lipid metabolism. The most common acyl-CoA esters in plants are 16:0-CoA, 18:2-CoA, and 18:3-CoA. Isothermal titration calorimetry (ITC) experiments were recently conducted to study the binding affinity of OsACBP1 and OsACBP2 for these three acyl-CoA esters. The results showed that the equilibrium dissociation constant (Kd) value of the binding activity between OsACBP1 and the three acyl-CoA esters was 0.031–2.36 mM, while the Kd value of the binding activity between OsACBP2 and the three acyl-CoA esters was 0.080–0.85 mM. OsACBP2 had a significantly higher binding affinity for unsaturated acyl-CoA than OsACBP1, while OsACBP1 and OsACBP2 exhibited the same binding activity with 16:0-CoA. This indicates that although OsACBP1 and OsACBP2 are up to 79% homologous, their functions in organisms are significantly different [30]. Therefore, it is of great interest to explore the biochemical properties of OsACBP to further understand its mechanism of action in *O. sativa*. Furthermore, since OsACBP1 and OsACBP2 both belong to the first class of ACBP family proteins, this research also revealed that the functions of ACBP family proteins from the same class might also be quite different.

### 3.3. The Function of ACBP Family Proteins in A. tatiana

*A. tatiana* contains six ACBP family proteins, which are the most intensively studied plant ACBP family proteins [3]. Recent studies have shown that knocking out AtACBP1 can enhance the tolerance of *A. tatiana* to freezing, while overexpression of AtACBP1 can enhance its sensitivity to freezing [31]. When AtACBP6 in *A. tatiana* was knocked out, the mutant was more sensitive to freezing than the wild type, while overexpression of AtACBP6 enhanced its tolerance to freezing [32]. This demonstrates that although AtACBP1 and AtACBP6 are both ACBP family proteins, their functions in *A. tatiana* against freezing are quite different. Studies have shown that AtACBP1 and AtACBP2 can bind heavy metal ions, thereby affecting their intracellular functions. Researchers reduced the tolerance of *A. tatiana* to Pb^2+^ by knocking out AtACBP1, while overexpression of AtACBP1 increased its tolerance to Pb^2+^ [33,34]. When overexpressing AtACBP2, *A. tatiana* significantly increased its tolerance to Cd^2+,^ and the protein alleviated the lipid peroxidation caused by heavy metals by participating in the repair of glycerophospholipids in the cell membrane [34,35]. AtACBP also regulates cellular physiology by binding lipids in the cell. For example, AtACBP1 can combine with very-long-chain acyl-CoAs (VLCACoAs) in the cell, and researchers found a significant decrease in very-long-chain fatty acids (VLCFAs) on cuticular waxes following the deletion of AtACBP1 [36]. AtACBP3 regulates the plant response to hypoxia by regulating the metabolism of VLCFAs [37]. Recent studies have found that AtACBP4 and AtACBP5 play important roles in seed germination and pollen development of *A. thaliana* [22,38]. AtACBP5 can also participate in the formation of the cuticle and help *A. thaliana* defend against pathogens and microbial infection [39].

## 4. The Function of ACBP Family Proteins in Mammals

### 4.1. The Influence of Ceramide Synthesis of ACBP

Ceramides, which are very crucial sphingolipids in cells, are important components of the cell structure and play a key role in regulating cell differentiation and growth [40]. There are six ceramide synthase (CerS) isoforms in mammals, and they are involved in the synthesis of ceramides with differing acyl chain lengths [41]. Studies have shown that acyl-CoA synthetase 5 (ACSL5) can catalyze the biosynthesis of long-chain acyl-CoA esters and influence ceramide biosynthesis in the cell through the interactions with CerS [42,43]. Recent studies have shown that by knocking out the ACBP gene in mice, the liver cytoplasm of mutant mice is unable to activate CerS3, while high-speed liver cytosol from wild type mice was able to activate CerS3. At the same time, the content of long- and very-long-chain ceramides in the mutant cells was significantly reduced [9]. These results indicate that ACBP plays an important role in ceramide synthesis. Therefore, to explore the process by which ACBP affects sphingolipid metabolism is of great significance for understanding human diseases that are triggered by sphingolipid metabolism.

### 4.2. The Influence of ACBP on Obesity

ACBP can modulate dietary behavior by regulating the glucose level and lipid metabolism in the body [44]. A recent study has shown that by inserting monoclonal anti-ACBP antibodies into mice to neutralize the extracellular pool of ACBP, the food intake of mice after starvation was reduced, and the plasma glucose levels were significantly increased. Furthermore, when tamoxifen (a method that is based on TAM induced ACBP excision) was injected into mice to remove ACBP in vivo, researchers found that the mice could still maintain normal blood glucose levels in the cells and that the metabolic changes in the plasma and brown adipose tissue were similar to those of normal mice after starvation [44]. Some research has reported that hunger causes the autophagy-dependent release of ACBP from cells, thereby leading to an increase in plasma ACBP concentrations. Neutralization of ACBP showed that this process abolishes the hyperphagia after starvation of mice. At the same time, ACBP in plasma is increased in obese people and mice compared to normal people and mice [45]. Recent results indicate that the translocation of ACBP from the intracellular to the extracellular space could cause feedback inhibition of autophagy. This demonstrates that the autophagy-associated release of ACBP elicits feedback inhibition of autophagy through different mechanisms [46]. Therefore, ACBP plays an important role in regulating the blood glucose level, lipid metabolism, and diet behavior in the body. Exploring its function will contribute to the development of new methods for treating obesity.

## 5. The Function of ACBP Family Proteins in Filamentous Fungi

### 5.1. ACBP from M. ruber Can Promote the Synthesis of Monascus Pigments

*M. ruber* is a traditional food fermentation strain with a long history of use in Asia [47,48]. MPs are an important synthetic product of *M. ruber* that have anti-inflammatory, anti-cancer, anti-atherosclerotic, anti-diabetic, and other functions [49]. There are two synthetic routes of MPs in *M. ruber*, namely synthesis by polyketide polymerase and fatty acid synthetase [50]. Researchers have recently expressed a fusion protein of the maltose-binding protein (MBP) and MrACBP in *E. coli* Rosetta DE3. The binding affinity of MrACBP was determined for different ligands using the microscale thermophotoresist (MST) binding assay. It was demonstrated that MrACBP in *M. ruber* had a high binding affinity for myristoyl-CoA, palmitoyl-CoA, and octyl-CoA, while the highest binding affinity was for myristoyl-CoA. Homologous overexpression of ACBP5 in *M. ruber* showed that MPs increased by 11.67%. MPs are synthesized by polyketide, which is generated by acyl-CoA [50]. It is speculated that overexpression of ACBP5 results in the expansion of the intracellular acyl-CoA pool, thereby improving this process. The results also demonstrate that the content of water-soluble pigment and ethanol-soluble pigment increased by 9.80%, and 12.70% via overexpression of ACBP5, which may be related to biomass. Therefore, MrACBP preferentially binds to myristoyl-COA and can affect MPs biosynthesis in the cell [13]. 

### 5.2. ACBP from A. oryzae Preferentially Binds to Long-Chain Acyl-CoA

*A. oryzae* is an important filamentous fungi and is widely used in food fermentation. *A. oryzae* is unable to produce mycotoxin; however, it can produce a variety of enzymes with biological activities, including protease, peptidase, and amylase, that are useful during the process of brewing soy sauce to increase its nutrition, flavor, and taste [51]; thus, it is considered as a safe strain for food fermentation [52]. Therefore, exploring the function of AoACBP in lipid metabolism of *A. oryzae* is conducive to developing different types of low-fat food. Recently, researchers fused enhanced green fluorescent protein (EGFP) and AoACBP to facilitate subcellular localization. The results showed that AoACBP is mainly located in the punctate structure of the cytoplasm, and it exhibits bidirectional mobility. Then the researchers treated *A. oryzae* with nocodazole, a microtubule-depolymerizing reagent, and the bidirectional motility of AoACBP disappeared after 10 min of treatment. When *A. oryzae* was treated with the actin defascinating reagent latrunculin B for 10 min, the bidirectional motility of AoACBP was still observable, suggesting that the bidirectional movement in punctate structures depends more on microtubules than actin [14].

To determine the biochemical properties of AoACBP in *A. oryzae* and its function in lipid metabolism, the AoACBP gene was cloned from *A. oryzae,* and the fusion protein of MBP and AoACBP were expressed and purified in *E. coli* cells. The activity of AoACBP with palmitoyl-CoA and myristoyl-CoA was analyzed using an MST assay. The results showed that AoACBP has a high binding affinity for palmitoyl-CoA and myristoyl-CoA, and the binding affinity for palmitoyl-CoA was the highest, indicating that AoACBP preferentially binds to long-chain acyl-CoA in *A. oryzae* [53].

### 5.3. Aoacb2 Is Secreted via the Unconventional Protein Secretion (UPS) Pathway in A. oryzae

By performing a BLAST search using the Acb1 gene sequence of *S. cerevisiae*, a recent study showed that *A. oryzae* has another ACBP family protein besides AoACBP, which was named Aoacb2. The researchers constructed a conditional strain with an HA tag attached to the C-terminus of Aoacb2 and expressed it to analyze the secretion of Aoacb2-HA in the medium by Western blot with an anti-HA antibody. To eliminate the interference of cell lysis, the researchers used EGFP connected with HA as the negative control and detected Aoacb2-HA and EGFP-HA in the supernatant of the carbon-starved culture. A small amount of EGFP-HA and a large amount of Aoacb2-HA were found in the supernatant, which indicates that cell lysis released part of EGFP-HA, while a large amount of Aoacb2-HA was secreted into the culture medium by *A. oryzae*. To verify whether AoACBP can also be secreted into the medium under the same conditions, the researchers also conducted an AoACBP secretion experiment. The results showed that AoACBP cannot be secreted into the medium under the same conditions. This indicates that carbon starvation induces AoAcb2 to be unconventionally secreted into the culture medium [54]. Therefore, two types of ACBP from *A. oryzae* undergo different secretion pathways. Understanding the detailed molecular mechanism of unconventional secretion may be advantageous for developing new biotechnological strategies to produce heterologous expression proteins. 

### 5.4. The Function of ACBP from Aspergillus flavus (A. flavus) May Affect Aflatoxin Production

*A. flavus* is a common plant and a human pathogen that can produce the carcinogenic aflatoxin. Aflatoxin is a secondary metabolite of *A. flavus,* and its biosynthesis is strongly influenced by various environmental factors [55]. Contamination of crops, such as cotton, peanuts, maize, and tree nuts by aflatoxin creates a substantial food safety risk, especially in developing countries [56]. Researchers used KEGG pathway analysis to investigate the expression levels of some proteins in *A. flavus* and demonstrated that the degradation of valine, leucine, and isoleucine in solid media can increase the synthesis of acyl-CoA compared to that in liquid media. Acyl-CoA, which is the precursor of aflatoxin, is important for aflatoxin synthesis [57]. Researchers cultured *A. flavus* in medium containing 2-phenylethanol and used GO Enrichment and KEGG pathway analysis to study the expression levels of some proteins in *A. flavus*. The results demonstrated that the related genes of aflatoxin, pyruvate, and propionate biosynthesis were significantly downregulated. Propionate is the precursor of acyl-CoA, so it may be the reduction in propionate leads to decreased acyl-CoA and aflatoxin [58]. Acyl-CoA is a critical cofactor in all organisms because it functions in numerous biosynthetic and energy-yielding metabolic pathways. Therefore, it is speculated that ACBP, as an acyl-CoA transport protein, may play a key role in aflatoxin synthesis [59].

## 6. The Function of ACBP Family Proteins in Yeast

### 6.1. ACBP Can Regulate the Synthesis and Degradation of NADPH in Yeast Yarrowia lipolytica

Among oil-producing microorganisms, NADPH is mainly produced through the pentose phosphate pathway, which plays an important role in intracellular lipid synthesis [60]. Recent research has shown that overexpression of NADP^+^-dependent glucose-6-phosphate dehydrogenase (encoded by ZWF1), an important enzyme in pentose photosphate pathway, results in a large reduction in intracellular biomass from 15.0 g/L to 9.5 g/L. It is suggested that the overexpression of ZWF1 may result in the imbalance of the NADPH/NADP^+^ content in the cell. When ACBP was overexpressed alone, the total amount of lipids in the cell increased from 9% to 25.1%. When ZWF1 and ACBP were overexpressed at the same time, the growth defects caused by the overexpression of ZWF1 alone were reversed, and a large amount of lipids were synthesized in the cells. It is speculated that overexpression of ACBP led to a large amount of acyl-CoA synthesis in the cells, leading to the oxidation of some NADPH, thus alleviating the imbalance of the NADPH/NADP^+^ content produced by ZWF1 overexpression. Therefore, ACBP is an important protein for intracellular lipid synthesis that can regulate the production and degradation of intracellular NADPH [61].

### 6.2. The function of ACBP from C. neoformans

C. neoformans is a human fungal pathogen that causes devastating cryptococcal meningitis, which threatens the lives of hundreds of thousands of people every year [62,63]. Recently, researchers examined the transcript levels of the ACB1 gene during bisexual mating and deleted the ACB1 gene in the hyper-filamentous serotype D strain XL280α and its congenic strain XL280a. The results show that Acb1 plays a crucial role in mating, and the acb1∆ mutant exhibited dramatically reduced filamentation relative to the wild type. Additionally, the introduction of the wild-type allele of ACB1 into the acb1∆ mutant restored the filamentation defect. Researchers made a Y80A mutated allele of Acb1 through site-directed mutagenesis and introduced it into the acb1∆ mutant. The results showed that the Y80A mutant could not restore the filamentation defect in the mutant [64]. The results also showed that the Y80 residue in the acyl-CoA binding domain is critical for the function of Acb1. Therefore, Acb1 regulates mycelium morphology by binding acyl-CoA, and this acyl-CoA binding ability is critical for the function of Acb1 during filamentation. Exploring the function of Acb1 can help to develop new methods to reduce the risk of pathogens.

The CnACBP protein from *C. neoformans* was expressed in *E. coli* BL21 (DE3) and purified using a Ni-NTA superflow column. The OCoA and model membranes were then added to the solution containing the target protein. The changes in the secondary structure of the protein in solution after the addition of OCoA and the model membranes were detected by Circular dichroism spectrometer (CD). The results showed that when OCoA was added to the protein solution, the CD spectrum of the protein did not change, indicating that the binding of OCoA had no effect on the secondary structure of CnACBP. When the model membranes containing 1-palmitoyl-2-oleoyl-sn-glycero-3-phosphoo-(1′-rac-glycerol) (POPG) were added to the protein solution, changes in the CnACBP CD spectra were detected, indicating that the secondary structure of the protein was affected by POPG addition. However, after addition of either the phospholipid 1-palmitoyl-2-oleoyl-sn-glycero-3-phosphocholine (POPC) or a mixture of POPG and POPC, the CD spectra of CnACBP did not change, indicating that POPC and the mixture of POPC and POPG had no effect on the CnACBP secondary structure. This indicates that CnACBP preferentially binds anion membranes over neutral membranes. By exploring the effects of OCoA and phospholipids on the structure and stability of CnACBP, it is helpful to understand the process of CnACBP transport to different parts of cells after binding with lipids [65]. 

### 6.3. Depletion or Overexpression of ACBP May Affect Intracellular Lipid Synthesis and Vacuole Function

An ACBP family protein is encoded by the Acb1 gene in yeast. Recently, studies have shown that through the depletion of the Acb1 gene, a large number of different shaped vesicles accumulated in the cell, and the structure of the plasma membrane also changed significantly. The multi-layered plasma membrane structures appeared in the plasma membrane [11,12]. When the researchers depleted the Acb1 gene, there was a significant accumulation of stearyl-COA in the cells and a significant decrease in myristoyl-COA, palmi-toleoyl-COA, and oleoyl-COA. This shows that ACBP plays an important role in intracellular lipid synthesis. After overexpression of the Acb1 gene, the lipid content in cells increased significantly, as did the acyl-CoA ester pool, indicating that ACBP can not only promote lipid synthesis but also the formation of an acyl-CoA ester pool [16,66].

Some research has shown that after depleting the Acb1 gene, the isolated vacuole cannot fuse with a homotypic vacuole, which may be because there is no ceramide in the vacuole after depleting the Acb1 gene, leading to the absence of SNARE proteins in the vacuolar membrane [12]. 

## 7. The Binding Characteristics of ACBP Family Proteins from Yeast and Filamentous Fungi

Some studies have shown that acyl-CoA esters have important physiological functions in the cell and that they play a key role in regulating ion channels, ion pumps, gene expression, and membrane fusion [43,67]. ACBP and its family of proteins bind to acyl-CoA esters through the ACBP domain and participate in different metabolic regulation of the body [2] (Table 1). Therefore, analysis of the binding relationship between the ACBP family proteins and acyl-CoA esters is helpful to further understand the important functions of ACBP in lipid metabolism. In filamentous fungi and yeast, ACBP and its family of proteins have a high binding affinity for acyl-CoA esters. Researchers analyzed the binding affinity of AoACBP from *A. oryzae* for nine kinds of acyl-CoA esters, including butyryl-COA, hexanoyl-COA, palmitoyl-COA, and myristoyl-COA. The results showed that the binding affinity of AoACBP towards palmitoyl-CoA, myristoyl-CoA, stearoyl-CoA, and eicosanoyl-CoA was significantly higher than that towards the other five acyl-CoA esters, indicating that AoACBP preferentially binds to long-chain acyl-CoA. The researchers applied the same method to determine the binding affinity of ScACBP from *S. cerevisiae* with nine different acyl-CoA esters. The results showed that ScACBP can bind C_4_–C_16_ acyl-CoA esters and that it has the highest binding affinity for palmitoyl-COA and myristoyl-COA. This indicates that when ScACBP binds C_4_–C_16_ acyl-CoA esters, long-chain acyl-CoA is preferred. However, ScACBP showed a lower binding affinity for stearyl-COA and eicosanoyl-COA, indicating that the binding affinity for very-long-chain acyl-CoA is lower [54,68]. The binding affinity of MrACBP of *M. ruber* for C_4_–C_22_ acyl-CoA was analyzed by MST assay and showed that MrACBP has a high binding affinity for myristoyl-COA, palmitoyl-COA, and octane-COA. The binding affinity for myristoyl-COA was the highest, indicating that MrACBP preferentially binds myristoyl-COA [13]. There are two ACBP family proteins in yeast *S. carlsbergensis*, among which ACBP type 1 exhibits a high binding affinity for palmitoyl-CoA [16]. Recently, a study using size exclusion chromatography (SEC) revealed that CnACBP from *C. neoformans* has a high binding affinity for oleoyl-COA and palmitoyl-COA, which can affect the spatial structure and stability of CnACBP [65] (Table 2).

## 8. Conclusions and Future Perspectives

In recent years, the research on ACBP and its family of proteins has increased, especially for mammals and plants. For example, HbACBP1 and HbACBP2 play an important role in latex production and lipid metabolism. However, the specific mechanism during the formation of latex by *H. brasiliensis* is still unclear and needs to be further explored [10]. Latex as an important commercial resource [26]; therefore, exploring the mechanism of HbACBP in latex formation lays an important foundation for developing high-quality latex and increasing latex production. By studying the structure and function of OsACBP1 and OsACBP2 from *O. sativa*, researchers demonstrated that although these proteins have high homology, there are significant differences in structure and function. Moreover, the characteristics of OsACBP1 and OsACBP2 for binding acyl-CoA esters are still unclear and require further exploration [30]. ACBP can interact with CerS in mice cells and affect ceramide and sphingolipid synthesis. However, the mechanism still needs to be further explored [9]. ACBP family proteins play an important role during lipid metabolism in the cell. Exploring the structure and physiological function of the ACBP family proteins is of great significance to improve the production of oil crops and explore the pathogenesis of diseases caused by lipid metabolic disorders.

Research of ACBP family proteins in filamentous fungi and fungal pathogens is relatively uncommon. The function of ACBP during lipid metabolism in filamentous fungi and its important role in physiological function have been confirmed. For example, the ACBP family proteins in *A. oryzae* and yeast can combine a variety of different acyl-CoA esters, which play an important role in the normal physiological function of the cell [68]. The unconventional secretion of ACBP exists in *A. oryzae*, yeast, and some mammals, but its potential mechanism remains to be explored [44,54,69]. ACBP in *M. rubers* can affect MPs synthesis; however, the mechanism remains to be explored [13]. *A. flavus* and *C. neoformans* are important pathogens, but there are few studies on ACBP function in them. Exploring the function of ACBP in fungal pathogens can contribute to the development of new methods to reduce the risk of pathogens. At present, most of the studies on ACBP family proteins in filamentous fungi are mainly focused on its functions in *A. oryzae* and *M. ruber*. However, the specific roles of the ACBP family proteins in lipid metabolism and their influence on the physiological functions of mycelium are rarely involved, and there is little related research on ACBP family proteins in other filamentous fungi [13,53,65,68]. There are two ACBP family proteins in *A. oryzae*. However, the current research on the function of ACBP in *A. oryzae* is limited to the binding affinity to its ligands. Research on the structure of ACBP family proteins and their effect on the physiological function of the cell is still lacking and needs to be further explored. As a food safety strain, *A. oryzae* has been used in the food fermentation for thousands of years. Exploring the structure and function of ACBP family proteins in *A. oryzae* lays an important foundation for further development of low-fat foods and further application of *A. oryzae* in the food industry [52].

## Figures and Tables

**Figure 1 jof-06-00034-f001:**
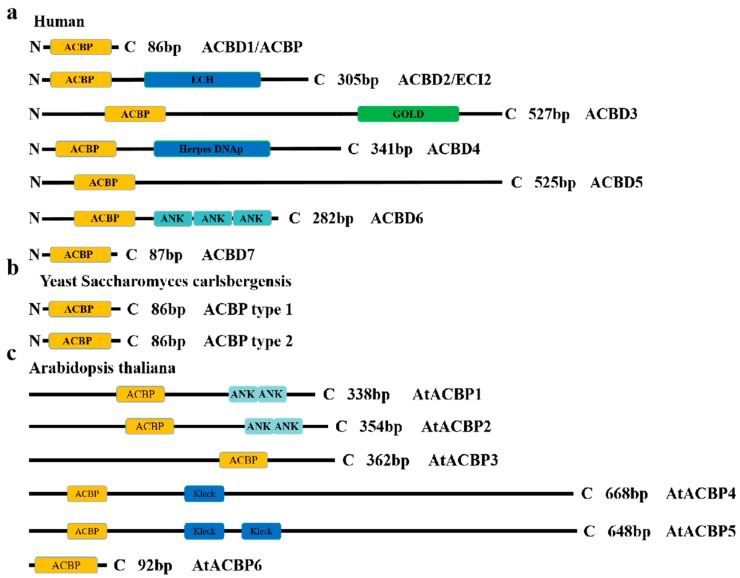
Schematic of key sequence domains of ACBP family proteins in humans, yeast *S. carlsbergensis*, and *A. thaliana*. All ACBP family proteins contain the ACBP domain, which binds to acyl-CoA ester. (**a**) There are seven kinds of ACBP family proteins in humans; (**b**) two kinds in yeast *S. carlsbergensis*; and (**c**) six kinds in *A. thaliana*.

**Table 1 jof-06-00034-t001:** The proposed function of Acyl-CoA-binding protein (ACBP) from filamentous fungi and yeast.

Fungus	ACBP Family Proteins	Proposed Function	Reference
*Aspergillus oryzae*	AoACBP	It localizes to punctate structures and preferentially binds to long-chain acyl-CoA	[14,54]
	Aoacb2	It is dispersed in the cytoplasm, important for the growth and undergoes unconventional secretion	[54]
*Aspergillus flavus*	ACBP	It may affect aflatoxin production	[57]
*Monascus ruber*	MrACBP	It improves the expansion of the intracellular acyl-CoA pool and increases MPs synthesis	[13]
*Saccharomyces cerevisiae*	Acb1p	It is required for normal vacuole function and could improve the formation of the acyl-CoA ester pool and ceramide	[12]
Yeast Yarrowia lipolytica	ACBP	It can improve acyl-CoA synthesis and regulates the production and degradation of intracellular NADPH	[61]
*Cryptococcus neoformans*	Acb1	It is important for mating and filamentation, and preferentially binds to anion lipid membranes	[64,65]
Saccharomyces carlsbergensis	ACBP type1/2	It can bind to acyl-CoA and improve the expansion of the intracellular acyl-CoA pool	[16]

**Table 2 jof-06-00034-t002:** Binding Characteristics of ACBP from Filamentous Fungi and Yeast with Acyl-CoA Esters.

Fungus	Proteins	Acyl-CoA Ester Binding	References
*Aspergillus oryzae*	AoACBP	C4:0 Butyryl-CoA	[57]
		C6:0 Hexanoyl-CoA	[57]
		C8:0 Octanoyl-CoA	[57]
		C10:0 Decanoyl-CoA	[57]
		C12:0 Dodecanoyl-CoA	[57]
		C14:0 myristoyl-CoA	[53,57]
		C16:0 palmitoyl-CoA	[53,57]
		C18:0 Stearoyl-CoA	[57]
		C20:0 Eicosanoyl-CoA	[57]
*Saccharomyces cerevisiae*	ScACBP	C4:0 Butyryl-CoA	[57]
		C6:0 Hexanoyl-CoA	[57]
		C8:0 Octanoyl-CoA	[57]
		C10:0 Decanoyl-CoA	[57]
		C12:0 Dodecanoyl-CoA	[57]
		C14:0 myristoyl-CoA	[57]
		C16:0 palmitoyl-CoA	[57]
		C18:0 Stearoyl-CoA	[57]
		C20:0 Eicosanoyl-CoA	[57]
*Monascus ruber*	MrACBP	C4:0 Butyryl-CoA	[14]
		C6:0 Hexanoyl-CoA	[14]
		C8:0 Octanoyl-CoA	[14]
		C10:0 Decanoyl-CoA	[14]
		C12:0 Dodecanoyl-CoA	[14]
		C14:0 myristoyl-CoA	[14]
		C16:0 palmitoyl-CoA	[14]
		C18:0 Stearoyl-CoA	[14]
		C20:0 Eicosanoyl-CoA	[14]
		C22:0 Acyl-CoA	[14]
*Saccharomyces carlsbergensis*	ACBP type 1	C16:0 palmitoyl-CoA	[16]
*Cryptococcus neoformans*	CnACBP	C18:1 oleoyl-CoA	[58]
		C16:0 palmitoyl-CoA	[58]

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
