# Peer review of "Advances in Understanding the Acyl-CoA-Binding Protein in Plants, Mammals, Yeast, and Filamentous Fungi"

_jof, 2020, doi:10.3390/jof6010034_

Round 1

Reviewer 1 Report

The Manuscript entitled “Advances in understanding of the acyl-CoA-binding protein in plants, mammals yeast, and filamentous fungi” brings attention to a conserved lipid binding protein. Overall the work is sound but needs some improvement before being accepted into the journal.

It is appreciated that this group chose to highlight an understudied topic, but the scope of the Journal of Fungi is topics related to fungi, and this special issue specifically focuses on lipids. There are large sections of this article that don’t have to do with fungi or lipids and may not be of interest to the target audience. With that being said, it is understandable that this protein may not be well studied in fungi. It is this reviewer’s opinion that the authors can do a better job relating the sections on mammalian and plant ACBP back to fungi and make the review a bit more focused, concentrating less on the details of plants and mammalian ABCDs and more on what the findings in mammals and plants could mean for fungi in regards to the homology of the proteins and conserved biological pathways.

Overall the organization of the article could also be improved. The section entitled “The homology of ACBP proteins” comes across as very confusing and repetitive, despite being an important section for understanding the rest of the paper.  There are also other sections where information from the introduction is almost copied and pasted into the body, while providing very little additional insight into the findings (e.g. data from Line 53 is repeated at Line 193).  There are certain sentences where the word “important” was repeated 2-3 times; it may be more valuable to use other descriptors, or to show how they are important by using literature. This article would be improved by focusing each paragraph so that there is a clear take home message from each section that is supported by recent literature. In its current state it read like a list of separate findings.

I think with some revision this article would make a valuable addition to the journal.

Author Response

Thank you very much for your valuable comments. Here is answers to the comments of the reviewers.

Reviewer #1:

  1. It is appreciated that this group chose to highlight an understudied topic, but the scope of the Journal of Fungi is topics related to fungi, and this special issue specifically focuses on lipids. There are large sections of this article that don’t have to do with fungi or lipids and may not be of interest to the target audience.

Answer: Yes, you are right. But the research of ACBP in fungi is less. I have added the research progress of ACBP in fungal pathogen, such as Aspergillus flavus and Cryptococcus neoformans.

  1. The section entitled “The homology of ACBP proteins” comes across as very confusing and repetitive, despite being an important section for understanding the rest of the paper.

Answer: Yes, you are right. I've changed the title to “General characteristics of structure and function of ACBP”. The content has been modified.

  1. There are also other sections where information from the introduction is almost copied and pasted into the body, while providing very little additional insight into the findings (e.g. data from Line 53 is repeated at Line 193).

Answer: The content of Line 193 has been changed and the more additional insight into the findings has been added.  

  1. There are certain sentences where the word “important” was repeated 2-3 times; it may be more valuable to use other descriptors, or to show how they are important by using literature.

Answer: Yes, you are right. The word “important” has been replaced by other descriptors.

  1. This article would be improved by focusing each paragraph so that there is a clear take home message from each section that is supported by recent literature.

Answer: After careful discussion by the author, the each paragraph has been focused and modified.

Reviewer 2 Report

  • Some of the introduction is choppy, could be written to flow better
  • Many of the organisms that will be discussed are pathogenic, the background should mention whether these proteins play a role in pathogenesis of these organisms.
  • The organization of the review is good; however, each subtopic is a specific review of the structure or function of one organisms’ ACBP…are there not more papers? For example: there are only two subtopics describing the function of ACBPs in mammals. It seems only very specific examples are reviewed. It would be nice to see a more general overview of each subtopic using numerous sources to give a broader overview of each topic instead of 1-2 very specific examples.
  • A table/figure showing the various functions of ACBP proteins would be good to have

Author Response

Thank you very much for your valuable comments. Here is answers to the comments of the reviewers.

Reviewer #2:

  1. Some of the introduction is choppy, could be written to flow better.

Answer: Yes, you are right. The introduction has been modified.

  1. Many of the organisms that will be discussed are pathogenic, the background should mention whether these proteins play a role in pathogenesis of these organisms.

Answer: The research of ACBP in fungal pathogen is less, I have added the research progress of ACBP in fungal pathogen, such as Aspergillus flavus and Cryptococcus neoformans, and the backgroud has been mentioned.

  1. The organization of the review is good; however, each subtopic is a specific review of the structure or function of one organisms’ ACBP…are there not more papers? For example: there are only two subtopics describing the function of ACBPs in mammals. It seems only very specific examples are reviewed. It would be nice to see a more general overview of each subtopic using numerous sources to give a broader overview of each topic instead of 1-2 very specific examples.

Answer: Yes, you are right. I have added some general overview to the subtopic.

  1. A table/figure showing the various functions of ACBP proteins would be good to have.

Answer: After careful discussion by the author, a table about ACBP various function has been added to the section of “7. The binding characteristics of ACBP family proteins from yeast and filamentous fungi”.

Round 2

Reviewer 1 Report

It is appreciated the effort that the authors have made to provide more information relating to fungi, and the new Table 1 really helps to bring more focus to the sections on Fungi. One critique is that the usage of "yeast" in the tables is not necessary in front of the scientific names of the fungi, as it is not used consistently and doesn't add to the tables. 

The other modifications to the manuscript have addressed the critiques from the initial comments, but in doing so have introduced many english errors that should be corrected prior to publication. The grammar and spelling in the new sections makes some of them hard to understand.

For example:

Line 182 "The influence of sbesity of ACBP" does not make sense. Maybe the authors could change this to "The influence of ACBP on obesity"

Line 197-206 the added sections have problems with verb tense "could causes" should be changed to "could cause" or "causes" and other verbs should follow along with the chosen tense. "of obese persons" should be changed to "in obese people"

Line 221-222 "MPs could biosynthesis via polyketide, and the acyl-CoA is the precursor of polyketide." It is unclear what the sentence here means.

Those are just a few examples, but there are many english errors that need to be corrected throughout the new sections, along with typo and spelling errors.

Author Response

Thank you very much for your valuable comments. Here is answers to the comments of the reviewers.

Reviewer #1:

  1. One critique is that the usage of "yeast" in the tables is not necessary in front of the scientific names of the fungi, as it is not used consistently and doesn't add to the tables.

Answer: Yes, you are right. After careful discussion by the author, the “yeast” in front of the scientific names of the fungi is deleted.

  1. Line 182 "The influence of sbesity of ACBP" does not make sense. Maybe the authors could change this to "The influence of ACBP on obesity".

Answer: Yes, you are right. I've changed the title to “The influence of ACBP on obesity”.

  1. Line 197-206 the added sections have problems with verb tense "could causes" should be changed to "could cause" or "causes" and other verbs should follow along with the chosen tense. "of obese persons" should be changed to "in obese people".

Answer: Yes, you are right. The content of Line 197-206 has been changed.

  1. Line 221-222 "MPs could biosynthesis via polyketide, and the acyl-CoA is the precursor of polyketide." It is unclear what the sentence here means.

Answer: After careful discussion by the author, the sentence has been changed to “MPs are synthesized by polyketide, which is generated by acyl-CoA”.

  1. Those are just a few examples, but there are many english errors that need to be corrected throughout the new sections, along with typo and spelling errors.

Answer: After careful discussion by the author, all the errors have been corrected.